# Using Ex Vivo Porcine Jejunum to Identify Membrane Transporter Substrates: A Screening Tool for Early—Stage Drug Development

**DOI:** 10.3390/biomedicines8090340

**Published:** 2020-09-10

**Authors:** Yvonne E. Arnold, Yogeshvar N. Kalia

**Affiliations:** 1School of Pharmaceutical Sciences, University of Geneva, Centre Médical Universitaire, 1 rue Michel Servet, 1211 Geneva, Switzerland; Yvonne.Arnold@unige.ch; 2Institute of Pharmaceutical Sciences of Western Switzerland, University of Geneva, Centre Médical Universitaire, 1 rue Michel Servet, 1211 Geneva, Switzerland

**Keywords:** intestinal drug efflux/uptake, ATP-binding cassette transporter, solute carrier transporter, ex vivo porcine intestine, Ussing chamber

## Abstract

Robust, predictive ex vivo/in vitro models to study intestinal drug absorption by passive and active transport mechanisms are scarce. Membrane transporters can significantly impact drug uptake and transporter-mediated drug–drug interactions can play a pivotal role in determining the drug safety profile. Here, the presence and activity of seven clinically relevant apical/basolateral drug transporters found in human jejunum were tested using ex vivo porcine intestine in a Ussing chamber system. Experiments using known substrates of peptide transporter 1 (PEPT1), organic anion transporting polypeptide (OATP2B1), organic cation transporter 1 (OCT1), P-glycoprotein (P-gp), breast cancer resistance protein (BCRP), multi drug resistance-associated protein 2 and 3 (MRP2 and MRP3), in the absence and presence of potent inhibitors, showed that there was a statistically significant change in apparent intestinal permeability *P_app,pig_* (cm/s) in the presence of the corresponding inhibitor. For MRP2, a transporter reportedly present at relatively low concentration, although *P_app,pig_* did not significantly change in the presence of the inhibitor, substrate deposition (*Q_DEP_*) in the intestinal tissue was significantly increased. The activity of the seven transport proteins was successfully demonstrated and the results provided insight into their apical/basolateral localization. In conclusion, the results suggest that studies using the porcine intestine/Ussing chamber system, which could easily be integrated into the drug development process, might enable the early-stage identification of new molecular entities that are substrates of membrane transporters.

## 1. Introduction

The human intestinal wall is a highly complex anatomical barrier with several challenging physiological functions. It must regulate the absorption of a wide variety of chemical entities with different physicochemical properties, which are vital for our well-being, but at the same time, prevent the entry of potentially harmful or toxic substances. Intestinal permselectivity is obviously extremely important, and a compromised intestinal wall leads to an increased risk of diseases in the intestinal tract and elsewhere in the body [1,2].

Many orally administered drugs are absorbed by passive transport; however, some are substrates of membrane transporters that facilitate uptake across the intestinal wall and entry into the hepatic portal vein, whereas others are eliminated by efflux transporters localized in the enterocytes [3]. Co-administration of drugs can result in transporter-mediated drug–drug interactions (DDI)—concurrent presence of transporter substrates and/or inhibitors can have a major effect on drug uptake and result in a highly variable oral bioavailability [4]. Therefore, identification of drug candidates that are transporter substrates at an early stage of drug development is crucial.

The need for an improved understanding of (i) the impact of membrane transporters on drug bioavailability (and hence efficacy) and (ii) transporter-mediated DDI is clear and numerous efforts have been made and are ongoing [5,6]. Today, more than 400 membrane transporters, divided into two superfamilies’ – the ATP-binding cassette (ABC) and the solute carrier (SLC) transporters-have been detected in the human genome [5]. It is thought that ~30 membrane transport proteins are involved in drug pharmacokinetics [7], and based on clinical evidence, some of them are considered to be clinically relevant [8,9], e.g., breast cancer resistance protein (BCRP) [10], peptide transporter 1 (PEPT-1) [11] organic cation transporter 1 (OCT1) [12] and organic anion transporting polypeptide 2B1 (OATP2B1) [13]. To date, much work has been done to study drug transporters in the liver and the kidney, but fewer studies have been performed on the intestine [14]. The availability of a highly predictive intestinal model would help (i) to promote our current understanding of transport processes mediated by efflux and uptake transporters, (ii) to improve knowledge regarding transporter mediated DDI and (iii) to bridge the gap between in vitro and in vivo studies.

Models capable of reliably predicting drug interactions with transporters are still scarce. Existing approaches include the use of in silico, in vitro, ex vivo, in situ and in vivo models [15]. Taking into consideration the anatomical and physiological complexity of the intestinal wall, the challenges in developing a reliable in vitro model are understandable. In vitro cell-based models, such as Caco-2, MDCK, HT29-MTX or Raji B cells have been used for many years to identify substrates for carrier-mediated drug uptake or efflux [16,17,18,19,20,21]. Such monolayer cell cultures are useful as a first screen, but the lack of a full physiological membrane, missing regional characteristics, e.g., variable distribution patterns of transport proteins in the different segments of the gastrointestinal tract [22], different transporter expression compared to the human intestine [23] and limited robustness in the presence of biorelevant media [24] strongly limit their predictive power. To better approach physiological conditions, ex vivo models using intestine from several animals, mostly rodents, have been tested [25,26,27,28,29]. A few experiments have studied drug permeability using human intestine [30,31,32]. Without doubt, viable human intestine would be the most appropriate tissue to study drug permeation in humans; however, its availability is limited. Porcine intestine is a good surrogate for human intestine, and the similarities between human and porcine intestine are well known [33,34,35]. To date, little work has been done to study intestinal drug permeation using porcine intestine [36,37,38,39,40,41,42,43,44]. We recently demonstrated that apparent permeability coefficients *P_app,pig_* measured using ex vivo porcine intestine in an Ussing chamber set-up showed a good correlation to effective permeability coefficients *P_eff,human_* reported in humans in vivo [45]. In addition to comparing passive transport results with those from human intestine, the study also demonstrated that P450 3A4 and P-gp activities were retained, confirming the viability of the tissue. Following those promising results, the aim of the present work was to evaluate the activity in porcine jejunum of a series of drug transporters that have been shown to be of clinical relevance in humans: this was done by quantifying the intestinal transport of known substrates in the absence and presence of specific inhibitors.

## 2. Materials and Methods

### 2.1. Chemicals

Digoxin, 95% and (+/−)-verapamil, 99% were obtained from Acros Organics (Morris Plains, NJ, USA); indomethacin, 98% and ranitidine hydrochloride were purchased from Alfa Aesar GmbH and Co KG (Karlsruhe, Germany), whereas losartan was obtained from Cayman Chemical (Ann Arbor, MI, USA). Atropine sodium salt, cefadroxil, fluvastatin natrium and valsartan were purchased from Sigma-Aldrich (St. Louis, MO, USA); rosuvastatin calcium salt was obtained from Santa Cruz Biotechnology Inc. (Dallas, TX, USA). Fexofenadine and sulfasalazine were purchased from Fluorochem Ltd. (Hadfield, UK), and rosiglitazone was obtained from GlaxoSmithKline (Brentford, UK). Agar, calcium chloride dihydrate, glucose hydrate, magnesium chloride hexahydrate, potassium chloride, sodium chloride, sodium phosphate monobasic and sodium hydrogen carbonate were obtained from Hänseler AG (Herisau, Switzerland).

### 2.2. Porcine Intestine

Fresh jejunum from female, 6 month-old Swiss noble pigs was harvested immediately after slaughter (Abattoir de Meinier; Meinier, Switzerland and Abattoir de Loëx; Bernex, Switzerland). The tissue was rinsed with ice-cold Krebs-bicarbonate ringer (KBR) buffer (120 mM NaCl, 5.5 mM KCl, 2.5 mM CaCl_2_, 1.2 mM MgCl_2_, 1.2 mM NaH_2_PO_4_, 20 mM NaHCO_3_ and 11 mM glucose; pH 7.4) [46] and then transported to the laboratory in ice-cold KBR under constant oxygenation with a gas mixture composed of 95% O_2_/5% CO_2_ (PanGas AG; Dagmersellen, Switzerland).

The jejunum was processed according to published protocols [28,47]. Briefly, the intestine was opened along the mesenteric border and carefully rinsed with ice-cold KBR. The tunica muscularis was removed, and the remaining tissue was sliced into pieces of approximately 1.5 cm^2^; tissue containing Peyer’s patches was eliminated.

### 2.3. Experimental Protocol for Permeation Studies

A six Ussing chamber system was used for the intestinal transport experiments (Physiologic Instruments; San Diego, CA, USA). The Ussing chambers were set up on a heating block connected to a VCC MC6 MultiChannel voltage–current clamp via six input modules, containing integral dummy membranes. A temperature of 38 °C (porcine body temperature) was maintained using a circulating water bath (ED-5, Julabo GmbH, Seelbach, Germany). The Ussing chamber system was set up following the protocol of Neirinckx et al. [28]. Briefly, each chamber contained two pairs of Ag/AgCl-electrodes (one pair for application/detection of currents, the other for detection of voltages) embedded in tips containing a mix of 3% agar in 3 M KCl. In a first step, the acceptor and donor compartments of the Ussing chamber were filled with preheated KBR, and any voltage difference between the electrodes and the electrical resistance due to the buffer solution was eliminated.

During all of the above procedures, both compartments were provided with a mix of 95% O_2_ and 5% CO_2_ for two reasons: first, to oxygenate the intestinal tissue and second, to circulate the buffer solution. Following this adjustment, the chambers were emptied and the intestinal tissue, mounted on sliders with an exposed surface area of 1.26 cm^2^, was inserted into the Ussing chambers, 45 min after having harvested the tissue from the animal. KBR was added to both compartments for an equilibration period of 30 min. In the next step, the buffer in the acceptor compartment was replaced with 7 mL fresh KBR. The experiment was started with the addition of either 7 mL KBR containing the substrate (100 µM, except for digoxin, 50 µM) or the substrate and the transporter specific inhibitor (100 µM, except in the case of P-gp inhibition, 50 µM verapamil) into the donor compartment. The choice of all substrates and inhibitors was based on the University of California, San Francisco—Food and Drug Administration (UCSF-FDA) TransPortal [48]. The substrates tested, their physicochemical properties and the inhibitors used are described in Table 1. Aliquots (400 µL) were withdrawn from the acceptor compartment every 20 min (t = 20, 40, 60, 80, 100 and 120 min—the end of the experiment), and the volume removed was immediately replaced with fresh, preheated buffer. The integrity of the intestinal tissue was monitored throughout the experiment by measurement of the transepithelial resistance (TEER). The minimal TEER of living, intact intestine was determined in previous experiments, and tissues with a TEER < 15 Ω·cm^2^ were eliminated and not used for the calculation of *P_app,pig_* [44]. At the end of the experiment, samples were taken from the acceptor and the donor compartment. The intestinal tissue was cut into little pieces and extracted for 6 h at room temperature in the mobile phase used for the UHPLC–MS/MS analysis (see Appendix A).

### 2.4. Analytical Methods

All samples were centrifuged for 10 min at 14,000 rpm using an Eppendorf Centrifuge 5804 (Vaudaux-Eppendorf AG, Schönenbuch, Switzerland), before being analyzed using a UHPLC–MS/MS system consisting of a Waters ACQUITY UPLC^®^ core system and a Waters XEVO™ TQ-S micro tandem quadrupole mass spectrometer (Milford, MA, USA). An ACQUITY UPLC^®^BEH C18 column, 1.7 µm, 25 × 2.1 mm connected to an ACQUITY UPLC^®^BEH C18 Vanguard™ pre-column, 1.7 µm, 5 × 2.1 mm was used for chromatographic separation. Mass spectrometry was performed in multiple reaction monitoring (MRM) mode. All details of the analytical methods used for each of the molecules tested can be found in the Appendix A.

### 2.5. Data Analysis

#### 2.5.1. Permeability Calculations

The apparent permeability coefficient *P_app,pig_* was calculated using the following equation:(1)$$P_{app,pig}=\frac{dc}{dt}\times \frac{V}{A\times C_{0}}\;\left(\frac{cm}{s}\right)$$
*dc*/*dt*: change in the acceptor concentration calculated from the slope of the concentration–time curve between 20 and 80 min*V*: volume of the buffer in the donor compartment (7 mL)*A*: exposed surface area (1.26 cm^2^)*C_0_*: initial concentration of the substrate in the donor compartment (100 µM, in case of digoxin: 50 µM)


#### 2.5.2. Calculating Drug Deposition (*Q_DEP_*) and Drug Permeation (*Q_PERM_*)

In addition to calculating *P_app,pig_*, the amounts of drug retained in the intestinal membrane (*Q_DEP_*) and present in the acceptor compartment (*Q_PERM_*) were also calculated using the following equations [65]:(2)$$Q_{DEP}=\frac{m_{int\;2h}}{m_{donor\;0h}}\times 100\;\left(\%\right)$$
(3)$$Q_{PERM}=\frac{m_{acc\;2h}}{m_{donor\;0h}}\times 100\;\left(\%\right)$$
*m_int 2h_*: amount of drug in the intestinal membrane at the end of the experiment (t = 2 h)*m_acc 2h_*: amount of drug permeated into the acceptor compartment at the end of the experiment (t = 2 h)*m_donor 0h_*: amount of drug in the donor compartment at the beginning of the experiment (t = 0 h)


### 2.6. Statistical Analysis

The results were presented as the mean ± SD. The statistical evaluation was performed using analysis of variance (one-way ANOVA) followed by Bonferroni’s multiple comparisons test or Student’s *t*-test. The significance level was fixed at α = 0.05.

## 3. Results and Discussion

Drozdzik et al. described the most clinically relevant multidrug transporters present in human intestine [22], and these have been compared (in a comprehensive study by Vaessen et al.) with the transporters reported in porcine jejunum [66]. With the exception of the organic cation transporter 3 (OCT3), which was below the limit of detection in porcine intestine, the remaining seven clinically relevant transporters were present and were tested in the present study (SLC transporters— peptide transporter 1 (PEPT1), OATP2B1 and OCT1; ABC transporters-P-gp, BCRP, MRP2 and MRP3). For each transporter *P_app,pig_* of a known substrate was determined in the absence and presence of an inhibitor.

### 3.1. SLC Transporters

Much less is known about the expression, regulation and function of SLC transporters and their importance for drug absorption and DDI than is the case for ABC transporters [4]. In the first part of this study, the presence and activity of three clinically important SLC transporters, namely PEPT1, OATP2B1 and OCT1, in ex vivo porcine intestine, was investigated using the Ussing chamber system.

#### 3.1.1. PEPT1

PEPT1, an H^+^ dependent apical uptake transport protein responsible for the absorption of oligopeptides and peptide-derived substances, was discovered in the early 1990s [67,68]. It plays an important role in the absorption of many peptide-derived drugs, such as antivirals, angiotensin-converting enzyme inhibitors or β-lactam antibiotics [69]. Cefadroxil, a β-lactam antibiotic and known substrate, was used to investigate PEPT1 activity in porcine jejunum [69,70]. The *P_app,pig,_* of cefadroxil, 2.82 ± 0.20 × 10^−6^ cm/s, was similar to, but ~1.8-fold lower than *P_app,rat_* (4.99 ± 0.50 × 10^−6^ cm/s) determined in the corresponding segment of the intestine [70]. Based on the transporter gene pattern, PEPT1 expression in the small intestine of rats has been reported to be similar to that in humans [71], whereas the PEPT1 concentration in porcine jejunum was reported to be higher [66]. A statistically significant decrease in *P_app,pig_* of cefadroxil in the presence of the PEPT1 inhibitor losartan [51] (*p* < 0.05) confirmed the presence and functionality of PEPT1 and its contribution to the absorption of cefadroxil across porcine jejunum ex vivo (Table 2 and Figure 1a).

#### 3.1.2. OATP2B1

In contrast to the PEPT1 transporter, the OATP uptake transporters have a broad substrate specificity [74,75,76]. Of the different OATP transporters, it was decided to investigate the activity of OATP2B1, since it exhibits the highest relative gene expression of the OATP family in the human intestine [65]. *P_app,pig_* of the OATP2B1 substrate rosuvastatin was 0.91 ± 0.64 × 10^−6^ cm/s (Table 2 and Figure 1(bi)), which is ~9-fold lower than the reported *P_app,human_* [30], although the OATP2B1 concentration in both tissues is similar [66]. There could be several possible reasons for this difference including, (i) the human intestine was taken from patients suffering from colonic cancer, compared to the intestine from healthy pigs used for our experiments, (ii) the intestinal wall is very susceptible to a high number of influences such as stress, diseases, intake of medications, age and nutrition [77,78]. However, a complete analysis of the different effects of these factors on the intestinal wall and their impact on intestinal drug permeation would require a separate study.

The *P_app,pig_* of rosuvastatin significantly increased in the presence of the OATP2B1 inhibitor rosiglitazone, indicating the successful inhibition of OATP2B1 (Table 2/Figure 1b(i)). Information about transporter localization of OATP2B1 in the intestinal wall is contradictory. Both apical and basolateral localization in human intestine have been described [79,80], and although Vaessen et al. detected the OATP2B1 in porcine jejunum, the localization was not specified [66]. Since the addition of rosiglitazone resulted in an increased permeability, OATP2B1 in the porcine intestine seemed to be located in the basolateral side of the intestinal wall. This result was confirmed by the Q*_DEP_* values. In the presence of rosiglitazone, Q*_DEP_* of rosuvastatin significantly decreased (Table 2 and Figure 1b(ii)), indicating the significantly reduced transport of rosuvastatin from the acceptor compartment back into the membrane via the OATP2B1 transporter.

#### 3.1.3. OCT1

OCT1 is a transporter involved in the uptake of cationic substances such as nutrients, endogenous amines and cationic drugs [12]. *P_app,pig_* of the OCT1 substrate ranitidine was in excellent agreement with *P_app,human_*, 5.07 ± 0.83 × 10^−6^ cm/s and 5.5 × 10^−6^ cm/s, respectively [72]. These results are around 5-fold lower compared to *P_eff,human_* (27.3 ± 24.7 × 10^−6^ cm/s), reported earlier [81]. As depicted in Table 2/Figure 1(ci), in the presence of the OCT1 inhibitor atropine, a significant decrease of *P_app,pig_* was observed. Many reports have pointed to the basolateral localization of the OCT1 transporter [57,82,83,84]. However, the apical localization of the transporter was reported in a study in 2013 [12]. The ranitidine results, which show the significant decrease of *P_app,pig_* and *Q_DEP_* (Table 2/Figure 1c) in the presence of atropine, suggest that OCT1 is localized on the apical side of the brush border membrane in the porcine intestine. Although Haslam et al. suggested that ranitidine is predominantly absorbed via the paracellular way [72], the fact that it was possible to significantly decrease *P_app,pig_* with the addition of an OCT1 inhibitor indicated the evident involvement of OCT1 in ranitidine permeation.

### 3.2. ABC Transporters

ABC transporters are responsible for the energy dependent efflux of a wide variety of substances, and they are often involved in drug resistance [85,86,87,88]. Here, we investigated the presence and activity of P-gp, BCRP, multi drug resistance-associated protein 2 and 3 (MRP2 and MRP3), respectively, in porcine jejunum.

#### 3.2.1. P-gp

P-gp is a transporter of high clinical relevance, and considerable information regarding its structure, DDI and drug resistance is available in the literature [89,90,91,92]. The activity and successful inhibition of P-gp in porcine intestine were already demonstrated in our previous work [45]. For two substrates, ranitidine (K_m_ = 0.27 mM, [93]) and cimetidine, P-gp inhibition resulted in a significant increase of *P_app,pig_* in the ileum, but not in the jejunum. These results were in line with the increase in P-gp concentration on going from the porcine jejunum to the ileum [94]. In this study, the aim was to test a substrate in the jejunum with higher P-gp affinity compared to the previously tested substances. Digoxin, a well-known cardiac glycoside, with a K_m_ = 0.73 µM was selected [95]. Due to the limited digoxin solubility in KBR at 38 °C, experiments were performed using a reduced substrate and inhibitor concentration—50 µM for each. The *P_app,pig_* in the jejunum was 0.38 ± 0.23 × 10^−6^ cm/s and lower than *P_app,human_* obtained in ex vivo human intestine—1.44 ± 0.72 × 10^−6^ cm/s (Table 2 and Figure 2a) [30]. Some possible reasons for the differences originating from the source of the human tissue are discussed in Section 3.1.2. However, an additional reason for this difference could be due to the structure variation between human and porcine P-gp. Even though 89% of the nucleotide sequences of human and porcine intestine is identical, porcine P-gp has only seven potential *N*-glycosylation sites compared to ten in human P-gp [94]. So far, the effect of the *N*-glycosylation sites on porcine P-gp function has not been resolved. Therefore, further studies have to be performed to clarify their function. *P_eff,human_* (30 × 10^−6^ cm/s) was around 80 times greater [96]; however, as mentioned in previous work, this difference is based on the different considerations of the surface areas [45,97]. In the presence of the P-gp inhibitor verapamil, *P_app,pig_* in the jejunum increased 4-fold to 1.64 ± 0.79 × 10^−6^ cm/s (Figure 2a). Compared to the results in our previous work, P-gp inhibition using verapamil significantly increased *P_app,pig_* of digoxin in the jejunum, possibly due to the higher P-gp affinity of digoxin compared to ranitidine.

#### 3.2.2. BCRP

BCRP is an apical efflux transporter with broad substrate specificity [98]. Here, sulfasalazine, a sulphonamide, was chosen as the substrate and fluvastatin as the inhibitor. As shown in Table 2 and Figure 2b, *P_app,pig_* of sulfasalazine was 0.006 ± 0.004 × 10^−6^ cm/s, which was ~10-fold less than *P_app,human_* [30]. The impact of the inhibition of viable BCRP transporter by fluvastatin on sulfasalazine transport was demonstrated by the 100-fold increase of *P_app,pig_* in the presence of the inhibitor.

#### 3.2.3. MRP2

Valsartan, a selective angiotensin II inhibitor, is a known substrate for the apical efflux transporter MRP2 [63]. As shown in Table 2 and Figure 3a, *P_app,pig_* was 1.2 ± 0.1 × 10^−6^ cm/s, which was 3- to 7-fold lower than *P_eff,rat_* [99,100], the only *P_eff_* value available in the literature. MRP2 was inhibited using indomethacin [101]. No significant difference of *P_app,pig_* of valsartan in the absence and presence of the inhibitor was found. However, *Q_DEP_* of valsartan was significantly higher in the presence of indomethacin, possibly indicating a successful inhibition of MRP2 (Figure 3b). Since the duration of the experiment was short and, given that the MRP2 concentration in porcine intestine is less than that of the other transporters tested [66], it was possible that although the duration of the experiment was sufficient for the inhibition of the transporter and to increase deposition of valsartan in the membrane, there was not enough time for valsartan to permeate into the acceptor compartment.

#### 3.2.4. MRP3

A second member of the MRP-family, MRP3, a basolateral efflux transport protein, was also studied using fexofenadine as substrate. *P_app,pig_* in the jejunum was 2.11 ± 0.73 × 10^−6^ cm/s (Table 3 and Figure 4a). Compared to *P_eff,human_*, the permeability was approximately five-fold lower [30]. Since fexofenadine is also a known P-gp substrate [102] and the activity of P-gp had already been shown, in a first step, P-gp was inhibited by the addition of verapamil, which resulted in a significantly increased *P_app,pig_* (Table 3 and Figure 4a). Most probably due to the reduced efflux, more fexofenadine was available for the basolaterally localized MRP3 transporter, accelerating the basolateral efflux of the API. This hypothesis was confirmed by the significantly reduced *Q_DEP_* in the presence of verapamil. Compared to *P_app,pig_* in the absence of any transporter inhibitors and *P_app,pig_* in the presence of the P-gp inhibitor, *P_app,pig_* significantly decreased in the presence of the MRP3 inhibitor indomethacin. *P_app,pig_* in the presence of both indomethacin (MRP3 inhibitor) and verapamil (P-gp inhibitor) was not statistically different, but interestingly, it was possible to significantly increase *Q_DEP_* in the presence of indomethacin and verapamil (Table 3 and Figure 4b). Although the use of indomethacin alone, implying a reduced basolateral efflux, did not change *Q_DEP_* significantly compared to *Q_DEP_* in the absence of any inhibitors, the additional inhibition of P-gp and the related reduced apical efflux of fexofenadine meant that *Q_DEP_* significantly increased.

### 3.3. Localization of Membrane Transporters in Porcine Jejunum

The quantitative data collected in the intestinal absorption experiments described above enabled a putative visualization of the localization of the different membrane transporters in porcine jejunum and a comparison with the observations in human intestine (Figure 5). Complementary visualization studies using immunohistochemistry and confocal microscopy are envisaged to provide further insight into the localization of the transport proteins.

## 4. Conclusions

The presence and activity of seven clinically relevant membrane transporters were confirmed in porcine intestine ex vivo by determining the uptake of known substrates in the absence and presence of specific inhibitors for each transporter. Different parameters—*P_app,pig_*, *Q_DEP_* and *Q_PERM_*—were used to evaluate the transport of each substrate; this enabled a more detailed interpretation of the transport data to be made. The results were interpreted where possible using published data on the expression of the transporters in different regions of the porcine intestine. The approach described here enables transporter substrates to be identified and studied using a healthy, easily accessible and viable physiological intestine, from a large omnivorous mammal, with a high similarity to human intestine. It has been shown that this model enables the evaluation of (i) passive transport in simple buffer solutions and biorelevant media and (ii) active transport of substrates of clinically relevant transporters and (iii) could be used to provide an insight into the interplay between transporters and enzymes for pre-systemic metabolism, e.g., CYP3A4. In a next step, even though a reliable, physiologically relevant evaluation of many important processes during drug permeation using ex vivo porcine intestine is already possible, the current model has to be extended to take into account the dynamic processes in the gastrointestinal tract, mimicking the passage through its different compartments. This will be an important step in approaching even more closely the highly complex physiological conditions and processes occurring during drug absorption in the human intestine.

## Figures and Tables

**Figure 1 biomedicines-08-00340-f001:**
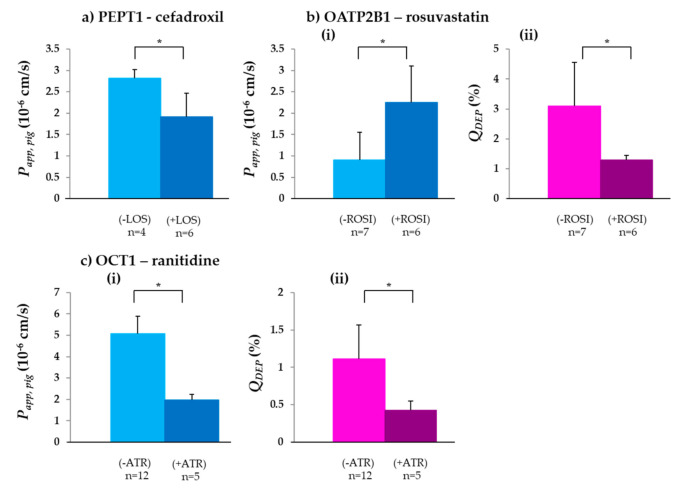
*P_app,pig_* and *Q_DEP_* values of three tested solute carrier (SLC) transporter substrates were significantly changed in the presence of the corresponding inhibitor: (**a**) *P_app,pig_* of the PEPT1 substrate cefadroxil in the absence (−LOS) and presence (+LOS) of the inhibitor losartan (LOS); (**b**) (**i**) *P_app,pig_* and (**b**) (**ii**) *Q_DEP_* of the OATP2B1 substrate rosuvastatin in the absence (–ROSI) and presence (+ROSI) of the OATP2B1 inhibitor rosiglitazone (ROSI); (**c**) (**i**) *P_app,pig_* and (**c**) (**ii**) *Q_DEP_* of the OCT1 substrate ranitidine in the absence (−ATR) and presence (+ATR) of the OCT1 inhibitor atropine (ATR) (mean ± SD; n = number of replicates). * *p* < 0.05.

**Figure 2 biomedicines-08-00340-f002:**
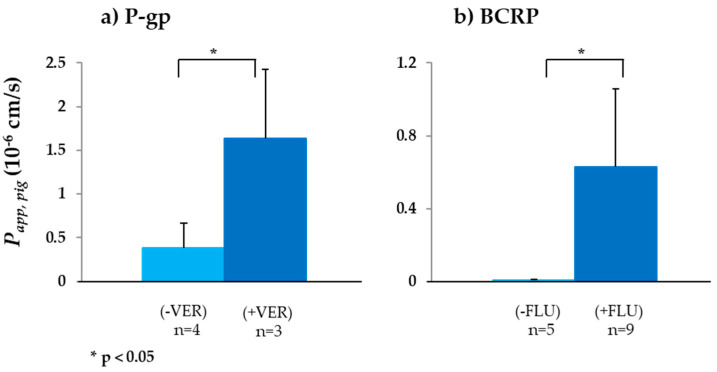
*P_app,pig_* values of ABC transport protein substrates: (**a**) *P_app,pig_* of the P-gp substrate digoxin in the absence (−VER) and presence (+VER) of the P-gp inhibitor verapamil (VER); (**b**) *P_app,pig_* of the BCRP substrate sulfasalazine in the absence (−FLU) and presence (+FLU) of the BCRP inhibitor fluvastatin (FLU) (mean ± SD; *n* = number of replicates).

**Figure 3 biomedicines-08-00340-f003:**
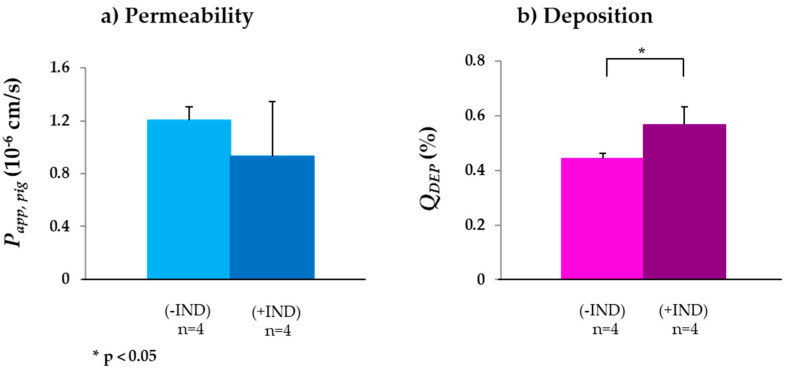
(**a**) *P_app,pig_* and (**b**) *Q_DEP_* of valsartan in absence (−IND) and presence (+IND) of the MRP2 inhibitor indomethacin (IND) (mean ± SD; *n* = number of replicates).

**Figure 4 biomedicines-08-00340-f004:**
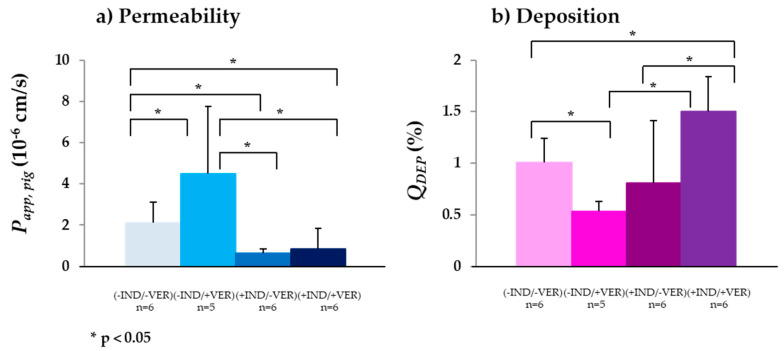
(**a**) *P_app pig_* and (**b**) *Q_DEP_* of fexofenadine in the absence (−IND)/(−VER) and presence (+IND)/(+VER) of the MRP3 inhibitors indomethacin (IND) and verapamil (VER) (mean ± SD; *n* = number of replicates).

**Figure 5 biomedicines-08-00340-f005:**
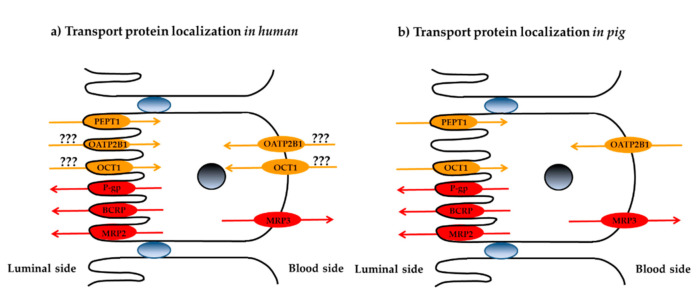
Comparison of the putative apical/basolateral localizations of membrane transporters in (**a**) human intestine and (**b**) porcine intestine. Uptake transporters are indicated in orange, efflux transporters are in red.

**Table 1 biomedicines-08-00340-t001:** Physicochemical properties of the drug molecules used to investigate the presence and activity of membrane transport proteins in porcine intestine using the Ussing chamber system (*n* = number of replicates).

Transport Protein	Substrate (BCS/BDDCS)	K_m_ (µM)	MW (g/mol)	log P [49]	log D ^a^ Octanol/H_2_O, pH 7.0	Solubility in KBR (n) (mg/mL)	Inhibitor (K_i_/IC_50_) (µM)
SLC Transport Proteins							
**PEPT1**	**Cefadroxil (-/III** [50]**)** **C_16_H_17_N_3_O_5_S**	n.d.	363.39	−0.4	−3.15	45.03 ± 1.10 (4)	Losartan (24/52) [51]
**OATP2B1**	**Rosuvastatin (III** [52]**/III** [50]**)** **C_22_H_28_FN_3_O_6_S**	2.4 [53]	481.54	0.13	−1.91	0.42 ± 0.16 (4)	Rosiglitazone (-/5.2) [54]
**OCT1**	**Ranitidine (III** [55]**/III** [50]**)** **C_14_H_22_N_2_O_3_**	70 [56]	314.40	0.27	−1.44	19.64 ± 2.78 (3)	Atropine (-/1.2) [57]
**ABC Transport Proteins**							
**P−gp**	**Digoxin (II** [55]**/IV** [50]**)** **C_41_H_64_O_14_**	73 [58]	780.94	1.26	1.29	0.04 ± 0.01 (3)	Verapamil (-/10.7) [59]
	**Fexofenadine (III** [55]**/III** [50]**)** **C_32_H_39_NO_4_**	n.d.	501.66	5.6	1.23	0.42 ± 0.16 (4)	
**BCRP**	**Sulfasalazine (IV** [60]**/II** [50]**)** **C_18_H_14_N_4_O_5_S**	0.7 [61]	398.39	0.4	−0.10	3.31 ± 0.14 (5)	Fluvastatin (5.43/-) [62]
**MRP2**	**Valsartan (III** [55]**/III** [50]**)** **C_24_H_29_N_5_O_3_**	30.4 [63]	435.5	3.68	−0.68	3.71 ± 0.68 (5)	Indomethacin (-/0.06) [64]
**MRP3**	**Fexofenadine (III** [55]**/III** [50]**)** **C_32_H_39_NO_4_**	n.d.	501.66	5.6	1.23	0.42 ± 0.16 (4)	Indomethacin (-/-)

Biopharmaceutical Classification System (BCS); Biopharmaceutical Drug Disposition Classification System (BDDCS); peptide transporter 1 (PEPT1), organic anion transporting polypeptide (OATP2B1), organic cation transporter 1 (OCT1), P-glycoprotein (P-gp), breast cancer resistance protein (BCRP), multi drug resistance-associated protein 2 and 3 (MRP2 and MRP3); ^a^ Values taken from SciFinder^®^; log D values were calculated using Advanced Chemistry Development (Software V11.02, ACD/Labs, Toronto, Canada).

**Table 2 biomedicines-08-00340-t002:** *P_app,pig_*, *Q_DEP_* and *Q_PERM_* of the tested transporter substrates in the absence and presence of the inhibitor, using ex vivo porcine jejunum and comparison with *P_app,rat_*/*P_app.human_* data from the literature.

Drug	Transporter	*P_app,pig_* (10^−6^ cm/s)		*Q_DEP_* (%)	*Q_PERM_* (n)(%)	*Q_DEP_* (%)	*Q_PERM_* (n)(%)	*P_app,rat_* (10^−6^ cm/s)	*P_app,human_* (10^−6^ cm/s)
		(−INH) (n)	(+INH) (n)	(−INH)		(+INH)		(−INH)	(−INH)
**Cefadroxil**	**PEPT1**	2.82 ± 0.20 (4)	1.91 ± 0.55 (6)	1.31 ± 0.40	0.50 ± 0.18 (4)	1.47 ± 0.25	0.31 ± 0.12 (6)	4.99 ± 0.50 [70]	-
**Rosuvastatin**	**OATP2B1**	0.91 ± 0.64 (7)	2.25 ± 0.85 (6)	3.10 ± 1.46	0.17 ± 0.09 (7)	1.30 ± 0.14	0.29 ± 0.16 (6)	-	6.95 ± 1.50 [30]
**Ranitidine**	**OCT1**	5.07 ± 0.83 (12)	1.96 ± 0.28 (5)	1.11 ± 0.46	0.62 ± 0.17 (12)	0.43 ± 0.12	0.21 ± 0.03 (5)	4.00 [72]	5.50 [72]
**Digoxin**	**P-gp**	0.38 ± 0.23 (4)	1.64 ± 0.79 (3)	0.62 ± 0.17	0.21 ± 0.12 (4)	0.40 ± 0.19	0.12 ± 0.04 (3)	6.4 ± 1.9 [14]	1.44 ± 0.72 [30]
**Sulfasalazine**	**BCRP**	0.01 ± 0.00 (5)	0.63 ± 0.43 (9)	3.23 ± 0.56	0.00 ± 0.00 (5)	2.47 ± 1.24	0.12 ± 0.07 (9)	2.76 ± 0.19 [73]	0.09 ± 0.06 [30]
**Valsartan**	**MRP2**	1.20 ± 0.10 (4)	0.93 ± 0.41 (4)	0.44 ± 0.02	0.17 ± 0.02 (4)	0.57 ± 0.07	0.24 ± 0.10 (4)	-	-

*P_app,pig_*: apparent intestinal permeability determined in this work using porcine intestine; *Q_DEP_*: amount of drug retained in the intestinal membrane at the end (*t* = 2 h) of the experiment; *Q_PERM_*: amount of drug in the acceptor compartment at the end (*t* = 2 h) of the experiment; *P_app,rat_*: apparent intestinal permeability determined using rat intestine (taken from literature); *P_app,human_*: apparent intestinal permeability determined using human intestine (taken from literature); *n*: number of replicates; (−INH): absence of inhibitor; (+INH): presence of inhibitor.

**Table 3 biomedicines-08-00340-t003:** *P_app,pig_*, *Q_DEP_* and *Q_PERM_* of fexofenadine without any inhibition (−IND/−VER), with inhibition of the P-gp transport protein (−IND/+VER), with the inhibition of the MRP3 transport protein (+IND/−VER) and with the inhibition of MRP3 and P-gp transport proteins (+IND/+VER), using ex vivo porcine intestine (*n* = number of replicates).

	(−IND/−VER) (*n*)	(−IND/+VER) (*n*)	(+IND/−VER) (*n*)	(+IND/+VER) (*n*)
***P_app,pig_* (10^−6^ cm/s) (*n*) ^a^**	2.11 ± 0.73 (6)	4.48 ± 3.29 (5)	0.64 ± 0.20 (6)	0.84 ± 0.52 (6)
***Q_DEP_* (%) (*n*)**	1.00 ± 0.24 (6)	0.54 ± 0.09 (5)	0.81 ± 0.61 (6)	1.50 ± 0.33 (6)
***Q_PERM_* (%) (*n*)**	0.06 ± 0.01 (6)	0.36 ± 0.20 (5)	0.27 ± 0.05 (6)	0.22 ± 0.10 (6)

^a^ number of replicates.

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
