# Peer review of "Using Ex Vivo Porcine Jejunum to Identify Membrane Transporter Substrates: A Screening Tool for Early—Stage Drug Development"

_biomedicines, 2020, doi:10.3390/biomedicines8090340_

Round 1

Reviewer 1 Report

This is an elegant study defining porcine intestinal drug transporters functional applying Ussig chamber assay (which is one of the established methods of functional transporter studied).

I have only some minor suggestions and comments:

In the introduction, I would suggest to clearly state that each functional segment of drug transporters possesses a specific batter of transporters (e.g. as referenced in Drozdzik et al.)

The results on OCT1 and OATP2B2, also contribute to the current discussion on membrane localization. Would it be possible to provide immumohistochemistry or confocal microspopy pictures with the membrane localization of these transporters?

Table 2 – all abbreviations should be explained in table legend

Author Response

This is an elegant study defining porcine intestinal drug transporters functional applying Ussig chamber assay (which is one of the established methods of functional transporter studied).

I have only some minor suggestions and comments:

In the introduction, I would suggest to clearly state that each functional segment of drug transporters possesses a specific batter of transporters (e.g. as referenced in Drozdzik et al.)

We thank the reviewer for the comment and have added the following text to line 63 of the revised manuscript.

“e.g. variable distribution patterns of transport proteins in the different segments of the gastrointestinal tract”

The results on OCT1 and OATP2B2, also contribute to the current discussion on membrane localization. Would it be possible to provide immumohistochemistry or confocal microspopy pictures with the membrane localization of these transporters?

We agree with the reviewer that imaging methods can help to identify transport protein localization in the membrane and that they would contribute to the ongoing discussion about the membrane localization of OCT1 and OATP2B1.

We will investigate this in complementary studies with colleagues from the histology department.  Although we have used confocal microscopy and immunohistochemistry techniques for studies with the skin, the focus of the present work was to use the quantitative impact of transport protein inhibitors on Papp, pig, QDEP and QPERM to confirm the presence of functional transport proteins in the intestinal tissue. The quantification of the effect also provided insight into the possible localization of the transport proteins and the results were compared to literature data.

The following text has been added to the manuscript (lines 330-332).

“Further studies using immunohistochemistry and confocal microscopy are envisaged to provide further insight into the localization of the transport proteins.”

Table 2 – all abbreviations should be explained in table legend

As proposed by the reviewer, we have added the missing explanations of the abbreviations used in Table 2 into the legend.

Reviewer 2 Report

Arnold and Kalia investigated in their study the vectorial transport of different probe compounds for well-established intestinal transporters by using the ex vivo model of the Ussing chamber and porcine jejunum.

The study addresses an important issue of drug development, namely the characterization of intestinal drug absorption. The used Ussing chamber experiment is well established and state-of-the-art. However, there are some major uncertainties which raise serious questions on the reliability and validity of the study, i.e. it remains uncertain whether both transport directions have been studied as usual and the validity of the used analytical method remains questionable. These data should be provided, otherwise sound and reliable conclusions cannot be derived. 

In detail, I have the following comments:

  1. Methods, point 2.2: The authors need to provide information about the applied time frames (between harvesting and start of the transport experiment, duration of the transport studies). This information is of great relevance as it determines the viability of the used tissue. The used TEER value is no valid parameter to assess tissue viability.
  2. Methods, point 2.3: How many experiments (different chambers) have been performed for each transporter and substrate? Have both transport directions (i.e. apical to basolateral and vice versa) been studied? This would be essential to conclude on the localization of a certain transporter. A simple apical to basolateral transport study would not be acceptable. This must be clarified and more detailed information should be added.
  3. Methods, point 2.3: The last sentence of the paragraph assumes that the mobile phase will be sufficient to lyse the tissue in order to determine the intracellular drug amount. I see two problems: 1. is the mobile phase (please mention its composition here) sufficient to lyse cells? and, 2. The applied extraction time of 6 h at 37 °C or room temperature suggests stability of all tested compounds at these conditions. I have doubts that all compounds will be stable enough. Please provide sufficient data proving this.
  4. Methods, point 2.4: Data on the validity of the analytical method should be provided. At least data on within- and between day accuracy and precision, matrix effects and stability should be provided (see the established guidelines by FDA and EMA on Bioanalytical Methods Validation). As all provided data are based on this method, these data are mandatory! Without them, the manuscript cannot be accepted. Alternatively, the authors may cite a paper in which they have described in detail their method validation.
  5. The authors should quantify the transporter proteins by appropriate methods such as targeted proteomics. Alternatively, gene expression may be used. This would convince the reader that the quality of tissue was comparable to previously published data from pig and that they are comparable to human intestinal transporter protein data.
  6. Assuming that the authors have investigated only the apical to basolateral transport, the OATP2B1 data do not make any sense assuming its apical localization. Inhibition of an uptake carrier should reduce and not increase the drug transport (permeability). Please explain.

Author Response

Arnold and Kalia investigated in their study the vectorial transport of different probe compounds for well-established intestinal transporters by using the ex vivo model of the Ussing chamber and porcine jejunum.

The study addresses an important issue of drug development, namely the characterization of intestinal drug absorption. The used Ussing chamber experiment is well established and state-of-the-art. However, there are some major uncertainties which raise serious questions on the reliability and validity of the study, i.e. it remains uncertain whether both transport directions have been studied as usual and the validity of the used analytical method remains questionable. These data should be provided, otherwise sound and reliable conclusions cannot be derived. 

In detail, I have the following comments:

  1. Methods, point 2.2: The authors need to provide information about the applied time frames (between harvesting and start of the transport experiment, duration of the transport studies). This information is of great relevance as it determines the viability of the used tissue. The used TEER value is no valid parameter to assess tissue viability.

Both timeframes are already mentioned in the manuscript (lines 118 & 132).

We have replaced “viability” by “integrity” which is the correct function of the TEER measurements and we thank the reviewer for the comment. We would like to specify that the TEER value mentioned in the text represents the threshold resistance value for determining tissue viability. Regarding the experiments performed using viable jejunum, the TEER average was around 40 Ω.cm2.

  1. Methods, point 2.3: How many experiments (different chambers) have been performed for each transporter and substrate? Have both transport directions (i.e. apical to basolateral and vice versa) been studied? This would be essential to conclude on the localization of a certain transporter. A simple apical to basolateral transport study would not be acceptable. This must be clarified and more detailed information should be added.

The number of experiments (different chambers) performed is specified by the term n in the Tables 2 and 3 as well as in the legend of the Figures 1 - 4.

In the case of cellular models, e.g. Caco-2 or MDCK, it is true that bidirectional studies are usually performed. Using cellular models, it has to be considered that there are major differences regarding the structure of the human intestinal wall, such as the lack of a full physiologic membrane, smaller pore size of tight junctions or different distribution patterns of transport proteins compared to the human intestine (Sun et al., 2008; Hilgendorf et al., 2000). Without any doubt, these models allow the identification of potential transport protein substrates (Pang et al., 2017), but the simulation of physiologic drug absorption including the most biopharmaceutically relevant processes remains challenging.

Using our experimental setup, the global aim is to develop a robust ex vivo model, mimicking the highly complex physiological intestinal absorption processes for orally taken drugs in vivo in humans. Under physiologic conditions, apical to basolateral direction is of most relevance. The intestinal mucosa is an anatomically complex asymmetric/heterogeneous biological tissue and therefore it made sense to conduct permeation studied from the “apical-to-basolateral” or “mucosal-to-serosal” direction – as would be the case in vivo. The asymmetrical structure is clearly apparent in the diagram and histological section shown below.                  

   PLEASE SEE ATTACHED RESPONSE

Given the anatomical asymmetry we evaluated transport exclusively in the apical-to-basolateral direction, i.e. Papp, a to b. The experiments in this work were the first experiments to evaluate the possibility to detect transport protein presence / activity simultaneously in porcine intestine ex vivo. Further experiments are planned to increase the complexity level similar to real conditions, e.g. to evaluate drug-drug interactions. Compared to the available literature, our methodology is in good agreement with already published work, where physiological membranes from different origins were used in the Ussing chamber and Papp was solely determined from the apical to the basolateral direction (Ungell et al., 1998; Neirinckx et al., 2010; Westerhout et al., 2014; Sjöberg et al., 2013; Miyake et al., 2018).

  1. Methods, point 2.3: The last sentence of the paragraph assumes that the mobile phase will be sufficient to lyse the tissue in order to determine the intracellular drug amount. I see two problems: 1. is the mobile phase (please mention its composition here) sufficient to lyse cells? and, 2. The applied extraction time of 6 h at 37 °C or room temperature suggests stability of all tested compounds at these conditions. I have doubts that all compounds will be stable enough. Please provide sufficient data proving this.

The extraction procedure was taken over from a well established technique used for skin permeation (Bachav et al., 2011; Kalaria et al., 2013; Kalaria et al., 2014), buccal mucosa and ocular tissue. Compared to the skin, intestinal tissue is an extremely fragile tissue. The presence of oxygen and nutrients are essential to keep the intestinal tissue viable (Nejdfors et al., 2000). Even in the Ussing chamber setup, where the intestinal tissue is surrounded by physiological buffer at pH 7.4 and the tissue is provided with oxygen as well as nutrients, we found first attritions of the intestinal epithelium after 120 min, avoiding a longer duration of the permeation experiment. The limited timeframe of the Ussing chamber setup is a general known disadvantage (Neirinckx et al., 2010; Westerhout et al., 2014; Polentarutti et al., 1999). Considering the very limited viability of the tissue in the experimental setup, the intestinal tissue, cut in little pieces, is for sure completely lysed in the extraction solution (no oxygenation, no nutrients) within 6 h.

The reviewer is asking to mention the compositions of the mobile phases in this paragraph. Since the compositions of the mobile phases varies between the substrates and to maintain continuity of the text, we decided to put the detailed compositions of the mobile phases into the Supplementary Information. If the Editor considers that Table S1 should be moved to the main text, this can of course be done.

The extraction was performed at room temperature (see line 138 in the text). The stability data of the compounds in the solvents were taken from literature. In contrast, no stability data of the compounds in KBR at 38°C were available in literature. Therefore, stability studies of the compounds under the mentioned conditions were determined. For the tested substances, no significant degradation was found. At the end of the permeation studies, mass balances were calculated, which can be used as an additional indicator for drug stability under the different conditions. With the exception of digoxin, the mass balances were within the range of 92 to 103%. In case of digoxin, the substance was stable in KBR, but the mass balance was ~ 89%. Further studies are planned to evaluate possible formation of dihydrodigoxin or digoxigenin.

  1. Methods, point 2.4: Data on the validity of the analytical method should be provided. At least data on within- and between day accuracy and precision, matrix effects and stability should be provided (see the established guidelines by FDA and EMA on Bioanalytical Methods Validation). As all provided data are based on this method, these data are mandatory! Without them, the manuscript cannot be accepted. Alternatively, the authors may cite a paper in which they have described in detail their method validation.

All of the analytical details including intraday accuracy and precision are presented in the Supplementary Information. We would add that according to ICH Guidelines, that intermediate precision / accuracy has to be done “depending on the circumstances under which the procedure is intended to be used”.  All analyses for a given experiment (one substrate in absence and presence of the corresponding inhibitor) were done on the same UHPLC-MS/MS system and on the same day; hence the focus on intraday precision/accuracy. Our samples consisted of compounds solubilized in Krebs-Bicarbonate Ringer buffer (120 mM NaCl, 5.5 mM KCl, 113 2.5 mM CaCl2, 1.2 mM MgCl2, 1.2 mM NaH2PO4, 20 mM NaHCO3, and 11 mM glucose; pH 7.4). As described in the article, before analysis all samples were centrifuged during 10 min at 14000 rpm and subsequently diluted with acetonitrile. Following this sample preparation protocol, no significant matrix effect was observed.

  1. The authors should quantify the transporter proteins by appropriate methods such as targeted proteomics. Alternatively, gene expression may be used. This would convince the reader that the quality of tissue was comparable to previously published data from pig and that they are comparable to human intestinal transporter protein data.

A quantification of the transport proteins in porcine intestine was already described in a comprehensive study by Vaessen et al (Regional expression levels of drug transporters and metabolizing enzymes along the pig and human intestinal tract and comparison with Caco-2 cells., Drug Metab. Dispos., 45 (2017) 353-360).  We are in discussion with colleagues to conduct a targeted proteomics study using intestine from the pigs used in the present study to see whether the results are consistent with those observed by Vaessen et al.

  1. Assuming that the authors have investigated only the apical to basolateral transport, the OATP2B1 data do not make any sense assuming its apical localization. Inhibition of an uptake carrier should reduce and not increase the drug transport (permeability). Please explain.

Indeed, only the apical to basolateral transport was evaluated, for further explanations see above. We agree that in the case of OATP2B1, the data would make no sense assuming apical localization. In the text, we brought up the possible basolateral localization of the transport protein – the hypothesis being supported by the QDEP values. We wrote:

“Since the addition of rosiglitazone resulted in an increased permeability, OATP2B1 in the porcine intestine seemed to be located in the basolateral side of the intestinal wall. This result was confirmed by the QDEP values.”

We intend to conduct immunohistochemistry studies and use confocal microscopy to try to provide visual evidence on the localization of the transporter proteins – this has been mentioned in the text (lines 330-332).

Thus, a final/definitive answer about the localization cannot yet be given. This topic is currently discussed in the field, since the transporter, assumed for a long time to be localized on the apical side, was recently found on the basolateral side of the human intestinal membrane (Keiser et al., 2017; Kobayashi et al., 2003).

References

Sun, H.; Chow, E.; Liu, S.; Du, Y.; Pang, K. The Caco-2 cell monolayer: usefulness and limitations. Expert Opin. Drug Metab. Toxicol. 2008, 4, 395-411, doi:10.1517/17425255.4.4.395.

Hilgendorf, C.; Spahn-Langguth, H.; Regardh, C.; Lipka, E.; Amidon, G.; Langguth, P. Caco-2 versus Caco-2/HT29-MTX co-cultured cell lines: permeabilities via diffusion, inside- and outside-directed carrier-mediated transport. J. Pharm. Sci. 2000, 89, 63-75, doi:10.1002/(SICI)1520-6017(200001)89:1<63::AID-JPS7>3.0.CO;2-6.

Pang, X.; Wang, L.; Kang, D.; Zhao, Y.; Song, W.; Liu, A.-L.; Du, G.-H. Effects of p-glycoprotein on the transport of DL0410, a potential multifunctional antiAlzheimer agent. Molecules 2017, 22, 1246, doi:10.3390/molecules22081246.

Ungell, A.-L.; Nylander, S.; Bergstrand, S.; Sjöberg, A.; Lennernäs, H. Membrane transport of drugs in different regions of the intestinal tract of the rat. J. Pharm. Sci. 1998, 87, 360-366, doi:10.1021/js970218s.

  1. Neirinckx, C. Vervaet, J. Michiels, S. De Smet, W. Van den Broeck, J.P. Remon, P. De Backer, S. Croubels, Feasibility of the Ussing chamber technique for the determination of in vitro jejunal permeability of passively absorbed compounds in different animal species, J. Vet. Pharmacol. Therap., 34 (2010) 290-297.
  2. Westerhout, A. van de Steeg, D. Grossouw, E.E. Zeijdner, C.A.M. Krul, M. Verwei, H.M. Wortelboer, A new approach to predict human intestinal absorption using porcine intestinal tissue and biorelevant matrices, Eur. J. Pharm. Sci., 63 (2014) 167-177.

Sjöberg, A.; Lutz, M.; Tannergren, C.; Wingolf, C.; Borde, A.; Ungell, A.-L. Comprehensive study on regional human intestinal permeability and prediction of fraction absorbed of drugs using the Ussing chamber technique. Eur. J. Pharm. Sci. 2013, 48, 166-180, doi:10.1016/j,ejps.2012.10.007.

Miyake, M.; Kondo, S.; Koga, T.; Yoda, N.; Nakazato, S.; Emoto, C.; Mukai, T.; Toguchi, H. Evaluation of intestinal metabolism and absorption using Ussing chamber system, equipped with intestinal tissue from rats and dogs. Eur. J. Pharm. Biopharm. 2018, 122, 49-53, doi:10.1016/j.ejpb.2017.09.015.

Y.G. Bachhav, A. Heinrich, Y.N. Kalia, Using laser microporation to improve transdermal delivery of diclofenac: increasing bioavailability and the range of therapeutic applications, Eur. J. Pharm. Biopharm., 78 (2011) 408-414.

D.R. Kalaria, P. Patel, V. Merino, V.B. Patravale, Y.N. Kalia, Controlled iontophoretic transport of huperzine A across skin in vitro and in vivo: effect of delivery conditions and comparison of pharmacokinetic models, Mol. Pharm., 10 (2013) 4322-4329.

D.R. Kalaria, P. Patel, V. Merino, V.B. Patravale, Y.N. Kalia, Controlled iontophoretic delivery of pramipexole: electrotransport kinetics in vitro and in vivo, Eur. J. Pharm. Biopharm., 88 (2014) 56-63.

  1. Nejdfors, M. Ekelund, B. Jeppson, B.R. Weström, Mucosal in vitro permeability in the intestinal tract of the pig, the rat, and man: species- and region-related differences, Scand. J. Gastroenterol., 35 (2000) 501-507.

B.I. Polentarutti, A.L. Peterson, A.K. Sjöberg, E.K.I. Anderberg, L.M. Utter, A.-L.B. Ungell, Evaluation of viability of excised rat intestinal segments in the Ussing chamber: investigation of morphology, electrical parameters, and permeability characteristics, Pharm. Res., 16 (1999) 446-454.

S.F.C. Vaessen, M.M.H. van Lipzig, R.H.H. Pieters, C.A.M. Krul, H.M. Wortelboer, A. van de Steeg, Regional expression levels of drug transporters and metabolizing enzymes along the pig and human intestinal tract and comparison with Caco-2 cells., Drug Metab. Dispos., 45 (2017) 353-360.

  1. Keiser, L. Kaltheuner, C. Wildberg, J. Müller, M. Grube, L.I. Partecke, C.-D. Heidecke, S. Oswald, The organic anion-transporting peptide 2B1 is localized in the basolateral membrane of the human jejunum and Caco-2 monolayers, J. Pharm. Sci., 106 (2017) 2657-2663.

Kobayashi, D.; Nozawa, T.; Imai, K.; Nezu, J.-I.; Tsui, A.; Tamai, I. Involvement of human organic anion transporting polypeptide OATP-B in pH-dependent transport across intestinal apical membrane. J. Pharmacol. Exp. Ther. 2003, 306, 703-708, doi:10.1124/jpet.103.051300.

Round 2

Reviewer 2 Report

The authors have addressed all comments and concerns made by the reviewer and modified their manuscript considerably. In addition, they have performed required additional experiments which improves the quality of the manuscript in a significant manner.

I have no further comment.